# Identification of potent high-affinity secondary nucleation inhibitors of Aβ42 aggregation from an ultra-large chemical library using deep docking

Michaela Brezinova [1], Z Faidon Brotzakis[1,2], Robert I Horne [1], Vaidehi Roy Chowdhury [1], Rebecca C Gregory[1], Yuqi Bian[1], Alicia González Díaz [1], Francesco Gentile [3,4] & Michele Vendruscolo [1]✉

## Abstract

**Alzheimer's disease is characterized by the aggregation of the Aβ peptide into amyloid fibrils. According to the amyloid hypothesis, pharmacologically targeting Aβ aggregation could result in disease-modifying treatments. The identification of inhibitors of Aβ aggregation, however, is complicated by complex technical challenges, which typically restrict to tens of thousands the number of compounds that can be screened in experimental aggregation assays. Here, we report a computational route to increase by 4 orders of magnitude the number of screenable compounds. We achieve this result by developing an open source pipeline version of the Deep Docking protocol, and illustrate its application to the discovery of secondary nucleation inhibitors of Aβ aggregation from an ultra-large chemical library of over 539 million compounds. The pipeline was used to prioritize 35 candidate compounds for in vitro testing in Aβ aggregation assays. We found that 19 of these compounds inhibit Aβ aggregation (54% hit rate). The two most potent compounds showed potency better than adapalene, a previously reported potent inhibitor of Aβ aggregation. Consistent with the intended mechanism of action, these two compounds also proved to be high-affinity binders of Aβ fibrils with an equilibrium dissociation constant in the low nanomolar range in surface plasmon resonance experiments. These results provide evidence that structure-based docking methods based on deep learning represent a cost-effective and rapid strategy to identify potent hits for drug development targeting protein misfolding diseases.**

**Keywords** Molecular Docking; Aβ42 Aggregation; Secondary Nucleation; Virtual Screening; Alzheimer's Disease
**Subject Categories** Neuroscience; Structural Biology

## Introduction

Many neurodegenerative diseases are associated with the process of protein misfolding and aggregation (Chiti & Dobson, 2017; Knowles et al, 2014; Hipp et al, 2019). In the case of Alzheimer's disease, one of the primary neuropathological features is the presence of extracellular deposits of the Aβ peptide, which are known as amyloid plaques (Hampel et al, 2021). Although the search for disease-modifying treatments for Alzheimer's disease has been particularly challenging (Cummings et al, 2023), the recent approvals by the Food and Drug Administration in the United States of three antibodies targeting Aβ has reinvigorated drug discovery efforts directed to the same target (Söderberg et al, 2022; Van Dyck et al, 2023; Mintun et al, 2021). Exhaustive experimental screenings for compounds that inhibit Aβ aggregation are, however, intractable, given the size of the chemical space of drug-like small molecules (estimated to be over $10^{60}$) (Bohacek et al, 1996; Dobson, 2004).

To increase the breadth of the accessible chemical space, virtual screening methods have established themselves as increasingly accurate alternatives (Kitchen et al, 2004; Shoichet, 2004; Schneider, 2010). Molecular docking is one such method (Meng et al, 2011; Gentile et al, 2022). The goal of molecular docking is to estimate whether two molecules (e.g., a protein target and a small molecule drug candidate) are likely to form a stable complex. Various large-scale docking campaigns have been conducted, finding potent binders (Bender et al, 2021), including a screening of 138 million molecules, discovering new chemotypes, among which was a subtype-selective agonist of the D4 dopamine receptor with affinity in the picomolar range (Lyu et al, 2019). However, even with molecular docking, screening large chemical libraries (over 1 billion) remains challenging. Docking 138 million molecules for one target in this campaign took 43,563 core hours with DOCK3.7, a licensed docking software, free for academic use (Coleman et al, 2013). Scaling this approach to a billion compounds and considering exclusive continuous access to 100 computing

[1]Centre for Misfolding Diseases, Yusuf Hamied Department of Chemistry, University of Cambridge, 12 Union Rd, Cambridge CB2 1EZ, United Kingdom. [2]Institute for Bioinnovation, Biomedical Sciences Research Center "Alexander Fleming", 34 Fleming Street, 16672 Vari, Greece. [3]Department of Chemistry and Biomolecular Sciences, University of Ottawa, 10 Marie Curie Pvt, Ottawa, K1N 6N5 Ontario, ON, Canada. [4]Ottawa Institute of Systems Biology, University of Ottawa, 451 Smyth Road, Ottawa, K1H 8M5 Ontario, ON, Canada. ✉E-mail: mv245@cam.ac.uk

cores, which is not standard for most research groups, the docking process would take 132 days. Furthermore, ligands would also have to be prepared in their 3D conformations, which requires significant extra time and access to sufficient hardware space.

With the advent of deep learning methods in structural biology (Jumper et al, 2021; Baek et al, 2021), there have been several successful attempts to accelerate and improve the docking process with the use of machine learning (Corso et al, 2022; Stärk et al, 2022; Wong et al, 2022; Méndez-Lucio et al, 2021; Lu et al, 2022; Zhou et al, 2023; Isert et al, 2023). Nonetheless, with faster and more precise docking methods, the problem of scaling to ever-increasing ultra-large library sizes still prevails. To address this problem, various active learning approaches have been described. Among these, pipelines applying conventional computational techniques are gaining popularity to screen a small subset of the library, using the results to train machine learning-based models to rapidly screen billions of molecules (Bedart et al, 2024). Docking a few million molecules compared to 1 billion can reduce the run time significantly. For example, in a similar set-up as described before, docking 5 million molecules should take only around 16 h, excluding the ligand preparation and other pipeline phases, opening up new avenues of express hit identification. One such pipeline is Deep Docking (Gentile et al, 2020), which enables ultra-large (in the order of billions) library screening up to 100-fold faster than traditional methods (Gentile et al, 2020). Depending on the available hardware and library size, the running of the pipeline should take 1–2 weeks. In this procedure, a randomly sampled subset of a library of available chemical space compounds is conventionally docked using a traditional physics-based docking software (for example, using FRED (McGann, 2011) or Glide (Friesner et al, 2004) to a target receptor. The resulting docking scores are then used to train a deep neural network, which in turn classifies hits vs non-hits for the whole ultra-large library, thus creating a dataset of predicted virtual hits. The training set is then augmented with a random sample of virtual hits, which are, again, conventionally docked and used to train a network that makes new, more accurate predictions for virtual hits. This active learning process is repeated for a set number of iterations until the dataset of virtual hits decreases to a size that can be conventionally docked and further processed.

Among other docking accelerator tools are MolPal (Graff et al, 2021), Lean Docking (Berenger et al, 2021), HASTEN (Kalliokoski, 2021), or NeuralDock (Sha et al, 2022). All these methods show promise in virtual hit enrichment and significant time reduction in docking ultra-large libraries compared to their main counterpart - traditional docking methods. Nonetheless, comparing them is difficult because of different internal structures, datasets, evaluation criteria, and a lack of experimental validation.

In this work, we build on the Deep Docking pipeline and make it fully open-source, exploring various possible optimizations. We then demonstrate and validate the use of the pipeline to discover small molecules targeting the amyloid fibrils formed by Aβ42, the 42-residue form of Aβ. This mechanism of action is motivated by the observation that the proliferation of Aβ is determined by an autocatalytic process in which existing amyloid fibrils promote the formation of new Aβ42 aggregates in a process known as secondary nucleation Aβ (Cohen et al, 2013). Therefore, blocking the catalytic sites on the surface of amyloid fibrils of Aβ should be highly effective against aggregation. This mechanism of action was

demonstrated for aducanumab (Linse et al, 2020), the first disease-modifying drug for Alzheimer's disease approved by the Food and Drug Administration (FDA) (Cummings et al, 2022). In our study, the resulting small molecule hits are further prioritized using in vitro assays to identify two potent inhibitors of Aβ42 aggregation. Notably, in a previous effort that prompted the current study, using an implementation of the structure-based search strategy described in this work based on conventional docking, rather than Deep Docking, we failed to identify inhibitors of Aβ42 aggregation, as the search space was limited to about 1.9 million compounds from the ZINC15 in-stock collection.

## Results

### Deep docking pipeline

In this work, we made the Deep Docking pipeline (Gentile et al, 2022) fully open-source. Our primary goal was not to improve the performance of the pipeline or to speed it up, but rather to make it available to a wider audience and demonstrate its use on a challenging target. The main changes required concerned the ligand preparation and docking phases, which relied on licensed software in the original workflow. The ligand conformations are now automatically retrieved from the ZINC20 website (Irwin et al, 2020) or generated using RDKit (Landrum, 2013) in SDF format. We note, however, that downloading conformations can be up to 405 times faster than RDKit-generating them on 1 core (using multiprocessing, this gap can be greatly reduced, however, still significant). This comparison is specifically for the energy-minimized conformation generation using RDKit. Subsequently, the conformations are docked with open-source AutoDock Vina (Trott & Olson, 2010) / Vina-GPU (Tang et al, 2022). These modifications are discussed in more detail in the section "Methods".

Briefly, the high-level protocol for the open-source pipeline is:

1. Randomly sample a set amount of molecules for training, validation, and a test set from the ultra-large library.
2. Prepare conformations for molecules to dock (download from ZINC20/generate using RDKit).
3. Dock ligands using Vina/Vina-GPU and extract docking scores.
4. Train a deep feed-forward neural network to distinguish hits from non-hits on training data, performing hyperparameter tuning on the validation set, and testing the performance on the test set.
5. Infer hit from non-hit status for the ultra-large library using the trained model.
6. Randomly sample a set amount of molecules from the inferred hits and enrich the training set.
7. Repeat steps 2–6 for a set number of iterations and then extract inferred hits for further downstream processing.

### Deep docking of Aβ42 fibrils

The open-source Deep Docking pipeline was run for five iterations to screen the ZINC20 library (Irwin et al, 2020), using as a target a recently reported structure of the amyloid fibril of Aβ42 (Xiao et al, 2015) and in particular targeting the low-solubility surface-exposed

binding site comprising residues [16]KVFAHLE[22] (see Fig. EV1), a putative secondary nucleation site on the fibrils, as done in previous studies for α-synuclein (Staats et al, 2023; Chia et al, 2022; Horne et al, 2024; Staats et al, 2020). The sizes of the initial training, validation, and test sets were 450,000 molecules each. In each subsequent iteration, the training set was enriched with 450,000 molecules randomly acquired from the potential hits of the previous iteration. Figure EV2 shows the number of predicted hits (A) as well as the ROC curves (B) over successive iterations. From the ROC curves, and the corresponding AUC values, it can be seen that the accuracy of the models increased at each iteration. Simultaneously, over iterations, the random samples of small molecules from the model-predicted hits were progressively more enriched with hits compared to the initial random sample (Fig. EV3). At the end of the pipeline, around 12 million virtual hits were predicted, with a cutoff Vina score of −7.23 kcal/mol in the last iteration.

## Downstream analysis and filtering

The predicted hits from the pipeline were further refined by retrieving the best-predicted 3 million molecules. Following the approach described in more detail in section "Methods", these small molecules were structurally clustered and the best-predicted-scoring molecule of each cluster was kept (651,245 molecules).

Subsequently, these molecules were docked to the target using Vina-GPU, and the best 100,000 scoring molecules in their best-predicted poses were afterwards scored with FRED, to make the further selection of hits more robust to artifacts of individual methods. After removing molecules with only their 2D representations available for docking (more discussed in section "Caveats"), the mean Vina score of shortlisted molecules was −6.9 ± 0.2 kcal/mol, and the mean FRED score was −4.0 ± 1.3. Despite the docking score being a relative number depending on the receptor itself and other properties, this is a significant improvement from the mean score of −5.1 ± 0.5 kcal/mol of a random sample of small molecules used in the first iteration. Since we did not optimize for

the FRED score, this score can be understood as a guideline for future filtering steps.

Using three different methods to select a subset of molecules based on their docking scores (as described in section "Downstream analysis"), a consensus of 978 molecules was reached. The mean Vina score of these molecules was −7.5 ± 0.2 kcal/mol, and the mean FRED score was −6.5 ± 0.6.

The consensus small molecules were then subjected to blood-brain barrier (BBB) permeability and suitability for the central nervous system (CNS) virtual testing, and 134 molecules were prioritized. Then, visual plot-based thresholding was used to further reduce the number of molecules to experimentally test to 59 (shown in Fig. 1). All these molecules are listed with their SMILES in Appendix Table S2.

## Experimental results

Out of 59 compounds obtained at the end of the deep docking pipeline, 35 were available for purchase and were experimentally tested in an initial aggregation assay, alongside adapalene, a previously reported inhibitor (Habchi et al, 2017) (Fig. 2). Full details on the identification of the compounds, as well as their normalized values of half-time aggregation, are available in Appendix Table S1. We found that 19 out of 35 compounds extended the half-time of aggregation by more than 50% over the negative control (i.e., the aggregation assay carried out in the absence of the compounds), resulting in a 54% hit rate. For these hits, the normalized half-time of aggregation was higher than 1.5. Molecules ZINC001678712262 (M1) and ZINC000730201302 (M11) were especially potent, with normalized half-times of about 9 and 7, respectively.

For the two most promising compounds (Fig. 3A,B), a full kinetic characterization was performed (Fig. 3C–G). This kinetic characterization was aimed at identifying the microscopic steps in the aggregation process of Aβ42 that were most affected by the compounds. Three different aggregation assays were carried out for this purpose: (1) unseeded aggregation, (2) low-seed aggregation (in the presence of 2% seeds, used to test secondary nucleation),

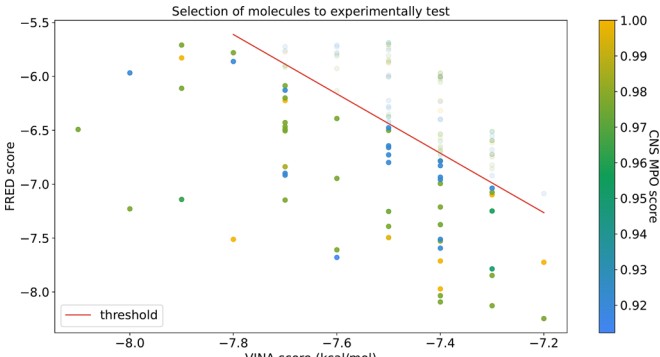

**Figure 1. Selection of the compounds for experimental validation.**

All prioritized molecules were required to pass the DeePred-BBB filter and have an MPO score above 0.9 (score ranging from 0-1). The compounds with both good FRED and Vina scores (below the red line) were preferred and formed the final selection. The prioritization of molecules below the red line was done only to reduce the number of compounds for experimental validation. Source data are available online for this figure.

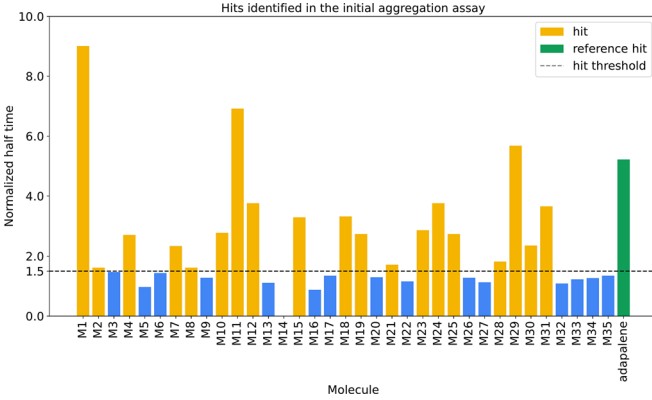

**Figure 2. Experimental validation of the 35 compounds selected using the deep docking pipeline from an initial library of 539 million compounds.**

We found that 19 of these 35 compounds inhibited Aβ42 aggregation, by extending the half-time of aggregation by over 50% (dotted line, representing 1.5 normalized half-time). Source data are available online for this figure.

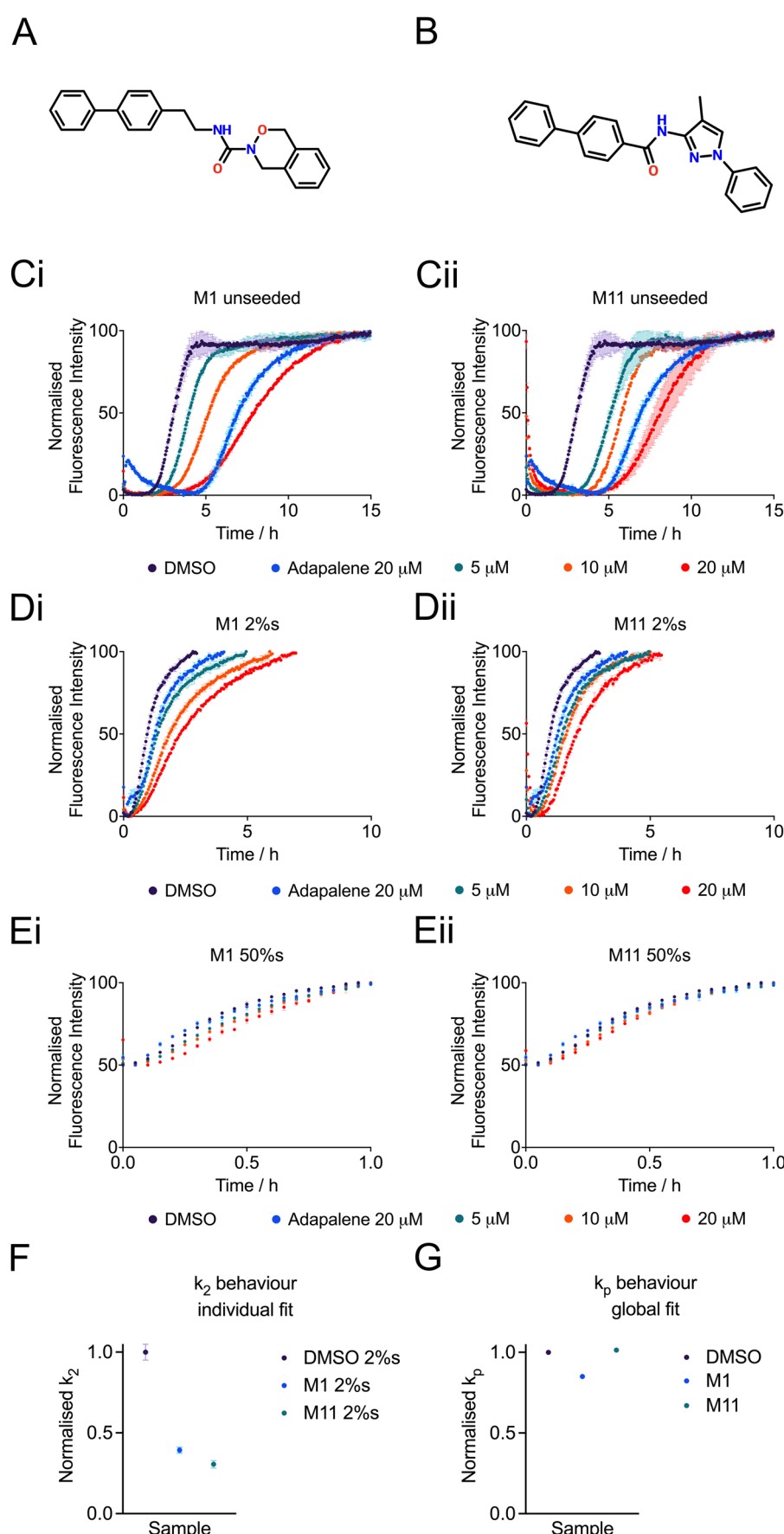

**Figure 3.   Detailed characterization of the effects on the aggregation kinetics of Aβ42 of the two most potent hits from the experimental screening of the 35 predicted compounds.**

(A, B) Structures of the two compounds: M1 - ZINC001678712262 (**A**), M11 - ZINC000730201302 (**B**). (**C–E**) ThT fluorescence traces for unseeded (**C**), low-seed (**D**), and high-seed (**E**) conditions. The negative control (Aβ42 aggregation in the absence of the compounds) is shown by a purple line. The positive control (adapalene (Habchi et al, 2017)) is shown by a blue line. (**F, G**) Fitting of the secondary nucleation - $k_2$ (**F**) and elongation - $k_p$ (**G**) rate constants for M1 and M11. Source data are available online for this figure.

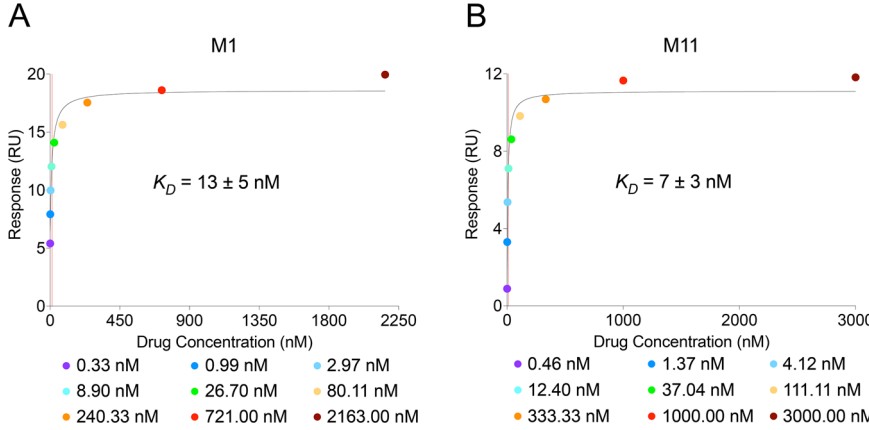

**Figure 4.   Determination of the binding affinity of the two most potent hits.**

Equilibrium analysis of SPR response at steady-state *vs* analyte concentration to determine the equilibrium binding affinity ($K_D$) of M1 (**A**) and M11 (**B**) to Aβ42 fibrils. The line trace shows the rectangular hyperbolic fit of the 1:1 binding model (Section Surface plasmon resonance (SPR) experiments) to the plotted data points shown in colored spheres. Source data are available online for this figure.

and high-seed aggregation (in the presence of 50% seeds, used to test elongation). The combination of these three assays enables the identification of the microscopic processes that are inhibited by the candidate compounds (Habchi et al, 2017). The results of these assays identified M1 as the best candidate, as it inhibited secondary nucleation rather well (Fig. 3Di).

We note that although we selected M1 to bind the surface of Aβ42 fibrils, hence blocking the catalytic sites for secondary nucleation, this compound also appeared to slightly inhibit elongation (Fig. 3G). This result could be expected since we did not explicitly optimize the conformational selectivity of the compounds. Previous strategies suggested that inhibiting secondary nucleation decreases the oligomer population (Chia et al, 2018). M11 performed equally well, with a milder effect on elongation, which was preferable (Fig. 3Eii). Both molecules seemed to have retained the potency seen in the initial experiment, performing better than adapalene (positive control, (Habchi et al, 2017)). Adapalene is an inhibitor of Aβ42 aggregation, (Habchi et al, 2017), however, via different mechanisms than fibril binding. M1 and M11 exhibit similar docking poses (Fig. EV4A,B), which differ from those of adapalene and ThT, as they bind a different groove on the fibril surface (Fig. EV4C,D). This is consistent with past observations that adapalene inhibits secondary nucleation through a mechanism that does not involve binding the amyloid fibrils (Habchi et al, 2017). This inhibition likely takes place through interactions with Aβ monomers or oligomeric intermediates, although this mechanism is typically less effective than direct inhibition at the fibril surface (Habchi et al, 2017; Linse et al, 2020). We also note that ThT is a potent binder of amyloid fibrils in

various binding modes, but it does not act as an aggregation inhibitor (Biancalana and Koide, 2010; Groenning, 2010).

With the kinetic characterization establishing the potency of M1 and M11 as inhibitors of surface-catalyzed secondary nucleation, the binding affinity of both molecules to Aβ42 fibrils was determined using surface plasmon resonance (SPR) spectroscopy. The SPR analysis indicated a strong interaction between the drug in solution and the surfaces of Aβ42 fibrils immobilized on a sensor chip (Figs. 4A,B and EV5). The equilibrium dissociation constant $K_D$ was determined by plotting the steady-state response against analyte concentration and fitting the resultant data to a 1:1 binding model. M11, which exclusively inhibited secondary nucleation in the aggregation assays, bound to Aβ42 fibrils with slightly higher affinity ($K_D = 7 \pm 3$ nM; $R^2 = 0.97$) compared to M1 ($K_D = 13 \pm 5$ nM; $R^2 = 0.96$).

We then assessed the ability of M1 and M11 to inhibit Aβ42 aggregation in a cellular environment using iPSC-derived glutamatergic neuronal cultures. After differentiating and maturing the neuronal cells for over 50 days, we modeled seeded aggregation by treating the cultures with 500 nM Aβ42 monomers and 50 nM fibrils, as previously described (Díaz et al, 2024). We administered four molecular equivalents of M1 or M11 alongside Aβ42. A two-treatment strategy was employed, where an initial protein and compound dose was followed by a second after 48 h. The culture was incubated to allow aggregation events to happen. Analysis at 24 and 48 h after the second treatment revealed that both M1 and M11 reduced the number of Aβ42 aggregates formed (Fig. 5A,B). These results are consistent with the in vitro mechanistic studies and suggest that M1 and M11 maintain their ability to inhibit secondary nucleation in a complex biological system.

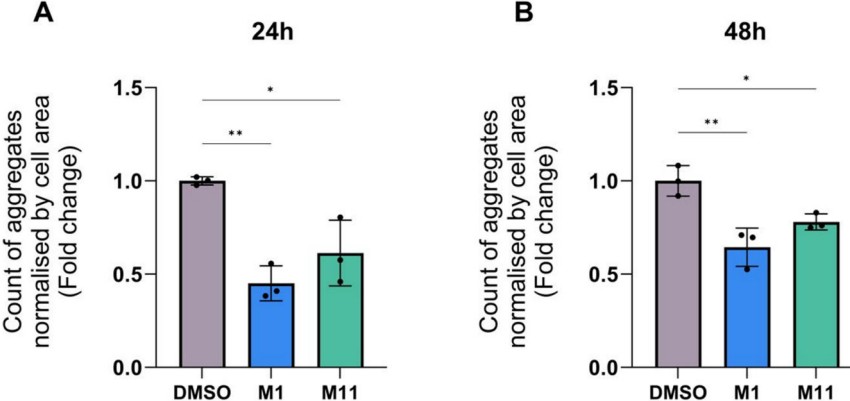

**Figure 5.  Inhibitory effects of M1 and M11 on Aβ42 aggregation in hiPSC-derived glutamatergic neurons.**

(A, B) Neurons were treated twice with 500 nM Aβ42, 50 nM fibrils, and DMSO or 2 μM of M1 or M11. Immunocytochemistry analysis of neurons was conducted at 24 and 48 h after the second treatment. The WO2 antibody was used to detect Aβ42. In the presence of M1 or M11, the number of Aβ42 aggregates formed was reduced as compared to the control. Data were presented as fold change in WO2-positive aggregate counts, normalized to cell area and relative to the monomer and fibril-treated control (DMSO). Data were mean ± s.e.m. across $n = 3$ technical replicates (wells) per condition, each quantified from 29 fields of view per replicate; results are from $N = 1$ independent experiments. Statistical significance was determined using a one-way ANOVA followed by Dunnett's test to correct for multiple comparisons (*$p < 0.033$, **$p < 0.002$, and ***$p < 0.001$). Source data are available online for this figure.

## Discussion

Starting the search from a chemical library of 539 million compounds, we developed and applied an open-source Deep Docking pipeline to select 59 potential inhibitors of Aβ42 aggregation for experimental validation. Among the 35 of these compounds that were commercially available, we identified 19 hits in our Aβ42 aggregation assay. We then took the two most potent compounds, ZINC001678712262 and ZINC000730201302, and carried out a detailed kinetic characterization, showing that they are indeed potent inhibitors of secondary nucleation, while also being fibril binders with a $K_D$ in the low nanomolar range. To our knowledge, no small molecule has hitherto been described to bind to Aβ42 fibril catalytic surfaces with such a high affinity. Additional studies on neuronal cultures showed that these molecules maintain their potency also in a complex cellular environment. These results open the possibility of blocking Aβ42 aggregation at the early stages when the fibril load in vivo is still relatively low. Virtual screening for BBB permeability, combined with the high affinity, also overcomes the problem of low efficacy often associated with the low bioavailability of small molecule drugs for CNS targets.

By running the deep docking pipeline, and by the subsequent experimental validation, we have illustrated the applicability of this procedure to a challenging problem in drug design for protein misfolding diseases. We were thus able to discover potent compounds outside of the traditional search space, which consists of smaller libraries of limited diversity. This pipeline can be integrated within larger drug discovery programs, followed by, for example, early hit optimization via searching for analogs in the same (or alternative) large chemical space and possibly finding ligands with optimal properties without any necessary custom synthesis (Sadybekov and Katritch, 2023).

To investigate in more detail the importance of screening ultra-large chemical libraries, we compared the most potent compounds, M1 and M11, with the compounds from the library of 1.9 million molecules that we previously screened, consisting of purchasable ZINC15 (Sterling and Irwin, 2015) (downloaded in 2020). The highest Tanimoto similarity values that we found were 0.48 and 0.50, respectively, showing that the chemical space containing M1 and M11 was originally not explored. Based on these observations, we suggested that the approach that we have reported is especially useful in the case of unconventional and challenging targets, such as Aβ, for which virtual screens at smaller scales can fail to identify hit compounds, as in our previous campaign.

As this method is general, we anticipate that it could be applied to other amyloid fibrils implicated in neurodegenerative diseases, including tau (Alzheimer's disease), huntingtin (Huntington's disease), TDP-43 (ALS), and α-synuclein (Parkinson's disease). These amyloid fibrils tend to have shallow binding pockets that make them challenging for drug discovery, and for which there are still not many known aggregation inhibitors. Various other targets, such as cancer-implicated proteins could be explored too.

Additional steps taken to adjust the Deep Docking pipeline make the whole process of small molecule discovery fully open-source, without dependence on any proprietary tools, representing a significant step toward the democratization of end-to-end drug discovery. This approach applies also to the downstream analysis, with the exception of using FRED as a complementary method to Vina, which can be replaced by a different open-source docking method in future studies.

Moreover, the alternative method to acquire 3D conformations provides a speed-up compared to traditional conformer generation methods. Utilization of Vina-GPU makes the docking much faster as well, compared to routinely used open-source programs such as AutoDock Vina, although it requires access to GPU computing resources. Our code hence provides an option to use AutoDock Vina instead, in cases where only CPU resources are available for docking.

We note that there are limitations to the method that we reported. Since it is aimed at accelerating docking screens, the

performance depends on the accuracy of the specific docking method adopted. As these tools need to be fast, they employ various approximations, which may lead to errors and a poor performance when comparing the binding affinity of the screened compounds (Bender et al, 2021). This is one of the reasons why consensus docking filters are usually incorporated to increase the predicted binding affinity accuracy, as it was done in the downstream analysis of this study (Palacio-Rodríguez et al, 2019). We should also mention that our goal was not to find all active ligands, but rather to identify a smaller set enriched in ligand candidates from a significantly larger library of non-binding molecules (Bender et al, 2021; Sadybekov and Katritch, 2023). At the same time, the pipeline is expected to perform well mainly for targets for which the binding prediction method works well. Hence, for targets whose structures are not known at high resolution or are of poor stability, this tool may not be expected to work well. Simultaneously, the search space also limits and biases the compounds that can be found. Despite the fact that chemical libraries of increasing size can significantly improve chances of finding hits, such chemical libraries tend to be less biased toward drug-like molecules than traditional in-stock subsets. Controlling for this aspect during the downstream analysis, as we illustrated in this work, could be useful in improving the probability of the molecules being effective drugs.

In conclusion, we have described the open-source deep docking pipeline and illustrated its application to target the process of aggregation of Aβ, a challenging problem for structure-based drug discovery. We thus identified 19 inhibitors of this process, with M1 (ZINC001678712262) and M11 (ZINC000730201302) being particularly potent. These results support the expectation that deep searches of the chemical space should lead to the effective identification of potent hit compounds. Importantly, it is broadly accepted that for challenging targets, the hit rates of virtual screening significantly deteriorate when larger libraries are investigated, due to artifact amplification (Lyu et al, 2019; Kuan et al, 2023). In this work, expanding the search to 539 million compounds led to a remarkable hit rate of 54%, while a previous virtual screening of 1.9 million compounds failed to retrieve any active compound. These findings thus motivate further investigations of the potential of ultra-large screens against targets currently undruggable by using conventional chemical libraries.

# Methods

### Reagents and tools table

| Reagent/Resource | Reference or Source | Identifier or Catalog or Product Number |
| --- | --- | --- |
| **Experimental models** | | |
| human induced pluripotent stem cell (hiPSC) line | RRID:CVCL_9S58, WT KOLF2 | WTSIi018-B-1 |
| **Recombinant DNA** | | |
| **Antibodies** | | |
| WO2 antibody | Sigma Aldrich | Cat #MABN10 |
| MAP2 antibody | Abcam | Cat #ab5392 |
| AlexaFluor555 anti-mouse | ThermoFisher | Cat #A-21422 |
| AlexaFluor488 anti-mouse | ThermoFisher | Cat # A-11039 |
| **Oligonucleotides and other sequence-based reagents** | | |
| **Chemicals, enzymes and other reagents** | | |
| Guanidine hydrochloride | MP Biomedicals | Cat #820540 |
| Sodium phosphate (dibasic) | Sigma-Aldrich | Cat #S0751 |
| Sodium phosphate (monobasic) | Sigma-Aldrich | Cat #S9390 or S3264 |
| Ethylene diamine tetraacetate | Fisher Chemical | Prod #10335460 |
| 96-well Half Area Black/Clear Flat Bottom Polystyrene NBS Microplate | Corning® | Prod #3881 |
| Dimethyl sulfoxide | Sigma-Aldrich | Cat #5.89569 |
| Tween-20 | Fisher Bioreagents | Prod #10485733 |
| Series S Sensor Chip CM3 | Cytiva | Cat #BR100536 |
| 1-ethyl-3-(3-dimethylaminopropyl)-carbodiimide (EDC) | Sigma-Aldrich | Prod #03450-5G |
| N-hydroxysuccinimide (NHS) | Sigma-Aldrich | Prod #56485-1G |
| Sodium acetate pH 4.0 | Cytiva | Cat #BR100349 |
| Ethanolamine-HCl pH 8.5 | Cytiva | Cat #BR100050 |
| N-(2-{[1,1'-biphenyl]-4-yl}ethyl)-3,4-dihydro1H-2,3-benzoxazine-3-carboxamide (M1) | Enamine | Cat #Z8478213987 |
| N-(4-methyl-1-phenyl-1H-pyrazol-3-yl)-[1,1'- biphenyl]-4-carboxamide (M11) | Enamine | Cat #Z1144421839 |
| Superdex Peptide 10/300 GL column | Cytiva | Cat #17517601 |
| Thioflavin T | Sigma-Aldrich | Prod #596200 |
| Sodium azide | Sigma-Aldrich | Prod #S2002 |
| GelTrex™ | Gibco | Cat #A141320 |
| mTeSR Plus | StemCell Technologies | Cat #100-0276 |
| ReLeSR™ | StemCell Technologies | Cat #100-0484 |
| StemPro Accutase | Gibco | Cat #A1110501 |
| DMEM/F-12 GlutaMAX | Gibco | Cat #10565018 |
| 1X N-2 | Gibco | Cat #17502048 |
| insulin | Sigma Aldrich | Cat #I9278 |
| nonessential amino acids | Gibco | Cat #11140050 |
| 2-mercaptoethanol | Sigma Aldrich | Cat # M6250 |
| Neurobasal | Gibco | Cat #21103049 |
| 1X B-27 | Gibco | Cat #17504044 |
| SB431542 | Sigma Aldrich | Cat #S4317 |
| LDN-193189 | Sigma Aldrich | Cat #SML0559 |
| PluriSTEM™ Dispase-II Solution | Sigma Aldrich | Cat #SCM133 |
| CHIR99021 | Sigma Aldrich | Cat #SML1046 |
| FGF2 | RnD Systems | Cat #3718-FB |
| Accutase | Sigma Aldrich | Cat #A6964 |
| ROCK inhibitor | Sigma Aldrich | Cat #SCM075 |
| penicillin and streptomycin | Gibco | Cat #15140122 |
| poly-L-ornithine | Sigma Aldrich | Cat #P4957 |

| laminin | Sigma Aldrich | Cat #L2020 |
|---|---|---|
| rhBDNF | R&D Systems | Cat #248-BDB |
| rhGDNF | R&D Systems | Cat #212-GD |
| L-ascorbic acid | Sigma Aldrich | Cat #A4544 |
| cAMP | Sigma Aldrich | Cat #D0627 |
| CultureOne | Gibco | Cat #A3320201 |
| DAPT | Tocris | Cat #2634/10 |
| BrainPhys™ Neuronal Medium | StemCell Technologies | Cat #05790 |
| SM1 | StemCell Technologies | Cat #05711 |
| PFA | Thermo Scientific | Cat #28906 |
| D-PBS ($+/+$) | Gibco | Cat #14040141 |
| Triton™ X-100 | Thermo Scientific | Cat #85111 |
| Immunofluorescence blocking buffer | Cell Signaling Technology | Cat #12411 |
| Hoechst 33342 | ThermoFisher | Cat #H3570 |
| **Software** | | |
| Prism 9 and 10 | GraphPad | |
| BIAevaluation | GE Healthcare | |
| Adobe Illustrator 2023 | Adobe Inc. | |
| **Other** | | |
| Biacore T200 | GE Healthcare | |
| ÄKTA pure | GE Healthcare | |
| FluoSTAR Omega plate-reader | BMG Labtech | |
| Superdex 75 10/300 GL column | GE Healthcare | |
| PerkinElmer CellCarrier Ultra 96 | Revvity | |
| Opera Phenix High-Content Confocal microscope | PerkinElmer | |
| Harmony High-Content Imaging and Analysis Software | Perkin Elmer | |

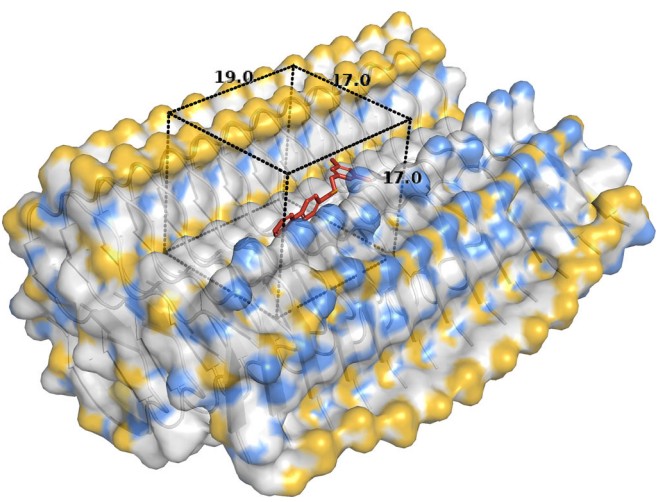

**Figure 6. Structure of the amyloid fibril of Aβ42 used as the target for the deep docking pipeline.**

An example ligand (red) in its binding site is shown inside the docking box (dotted black line). The figure is created using PyMOL (Schrödinger LLC, 2021). Source data are available online for this figure.

## Methods and protocols

### Screening data preparation

**Ligands.** Similar to the original protocol (Gentile et al, 2022), an ultra-large library of ~1 billion small molecules from the ZINC20 database (Irwin et al, 2020) was used. The ZINC20 database is publicly available and aggregates commercially available and annotated compounds. The library was available in a pre-processed version for machine learning applications, with enumerated isomeric configurations and dominant ionization states for each molecule as well as Morgan circular fingerprints of 1024 bits and radius 2 (Gentile et al, 2022). The dataset was filtered by the molecular weight ($\leq 360$ Da), as this has been defined as a more desirable range for molecules for the central nervous system (Wager et al, 2010), and allowed us to reduce the size of the chemical space to be searched. After the filtering, we were left with a subset of 539 million molecules that formed our final library to screen. Our code repository provides a script to run such library filtering as well as filtering on other parameters, including the upper bound for log$P$ (lipophilicity), lower

and upper bound for TPSA (topological polar surface area), and upper bound for HBD (hydrogen bond donor) number.

**Target structure.** For the case study of an amyloid fibril of Aβ42 as a target structure, a pre-processed version of the PDB DOI: 10.2210/pdb2MXU/pdb was used. The target structure and corresponding binding site are shown in Fig. 6. We used Fpocket (Le Guilloux et al, 2009), a volume-based software for the detection of binding pockets for small molecules, to identify multiple potential binding sites Fig. EV1A. Since our aim was to inhibit the secondary nucleation step in Aβ42 aggregation (Cohen et al, 2013), we narrowed down the search to the exposed sites on the surface. Next, we added a solubility filter using the structurally-corrected version of CamSol (Sormanni et al, 2015), a computational method to identify regions of low solubility on the surfaces of protein structures (Fig. EV1B). This analysis resulted in the choice of a binding site comprising residues [16]KVFAHLE[22] (Fig. 6).

### Hardware

The pipeline was run on a high-performance computing server of the University of Cambridge (CSD3). This server provides $\geq 1$ TB of allocated space and access to CPU cores as well as NVIDIA A100-SXM-80GB GPU cores. Hardware requirements for the pipeline are detailed more in the original protocol (Gentile et al, 2022). However, the full library of SMILES and Morgan fingerprints has a size of around 267GB, hence this amount of disk space is recommended, along with additional space for intermediate files and results.

### Open-source deep docking pipeline outline

The open-source pipeline was built on the blueprint of the deep docking pipeline, described in more detail in the original protocol (Gentile et al, 2022), with the main changes highlighted in Fig. 7, primarily in the Phase 2—Ligand preparation and Phase 3—Conventional docking phases. Briefly, the original pipeline, and

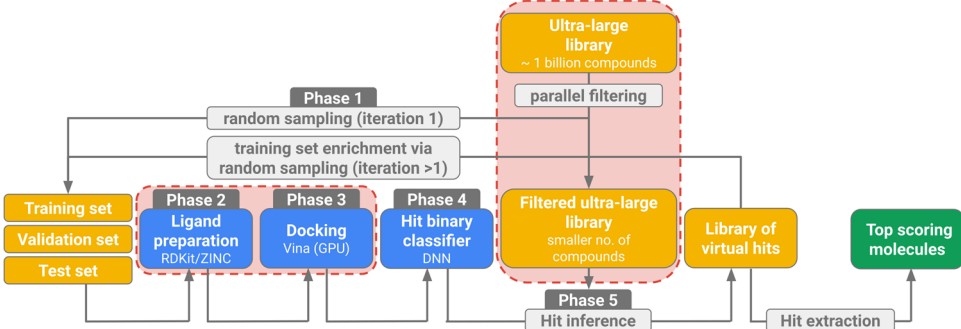

**Figure 7. Overview of the deep docking pipeline.**

The parts highlighted in red represent modifications introduced in the original protocol.

thus our pipeline as well, uses a deep feed-forward neural network as a binary classifier to perform hit identification, using Morgan fingerprints representations as input embeddings. Before training, the docking scores are pre-processed to assign binary labels (hits and non-hits) based on a threshold. This threshold is calculated using user-defined parameters and the affinity values of the top-scoring molecules. These parameters specify the percentage of compounds to be considered hits in the first and last iterations, with the percentage for all other iterations determined through linear interpolation. The threshold should get more strict with each iteration. In each iteration, hyperparameter tuning is conducted to determine the optimal settings for learning rate, batch size, number of hidden layers, number of nodes, dropout rate, class weights, and oversampling. This is done from 24 predefined hyperparameter configurations.

The selected testing, validation, and initial training set sizes were 450,000, and the protocol was run for five iterations. The size of datasets and the number of iterations were chosen to provide a balance between performance and computational cost, as suggested in the original protocol paper. In each subsequent iteration, the training set was enriched by 450,000 predicted hit compounds, randomly sampled from the results of the previous iteration's inference. All other parameters were kept as default in the original protocol—the percentage of top molecules considered as virtual hits in the first iteration to be 1%, the percentage of top molecules considered as virtual hits in the last iteration 0.01%, and the recall value to be 0.9. At the end of the pipeline, the top 3 million molecules (based on the neural network predictions) were selected for further downstream analysis.

As this protocol uses Vina docking instead of FRED or Glide, and the precision of Vina is only one decimal place, the code was slightly modified to accommodate this during the acquisition of labeled data following the docking phase.

Phase 2—Ligand preparation. The ligand preparation phase included preparing the conformations of the pre-sampled molecules for conventional docking onto the binding site. As AutoDock Vina and/or Vina-GPU were chosen for docking, which perform conformation sampling and pose generation by default, a single 3D conformation per ligand was required. Instead of generating conformations based on the SMILES representations, an alternative method of downloading 3D structures in SDF format directly from

the ZINC20 library was utilized instead. This strategy offers a significant speed-up and reduction in resources. On 1 core, the generation of 200 low-energy ligand conformations (of mean weight 334 Da) with RDKit takes on average around 142 min, while downloading the same number of ligands takes around 21 seconds. It is worth noting that the generation code is optimized for multiprocessing use and hence it is advised to use more cores in this process and reduce generation time (for example, to 22 min on 10 cores). This comparison is specifically for energy-minimized conformation generation using RDKit. Structure download was achieved using custom scripts performing parallel curl download requests on the ZINC20 database. Currently, each script is set to download a maximum of 200 molecular structures to prevent server time-out issues. As the structure downloaded from ZINC20 might sometimes correspond to a different configuration of the molecule than the one stored in our SMILES database, where necessary, Morgan fingerprint representations were adjusted to match the 3D structure. For molecules associated with multiple stereoisomers that were not available in the ZINC20 original database (i.e., whose stereochemistry was not entirely specified), RDKit was used to generate low-energy conformers using an adaptation of the code from Ebejer et al (Ebejer et al, 2012). After the generation, in both methods, molecules were converted to PDBQT format used by AutoDock Vina/Vina-GPU via OpenBabel (O'Boyle et al, 2011).

Phase 3—Conventional docking. This phase encompassed the conventional docking process of the sampled subset of the library that is used to generate labeled data for training the neural network. For the docking, an open-source GPU version of AutoDock Vina was chosen, Vina-GPU (Tang et al, 2022), with a reported average speed-up of 21 compared to AutoDock Vina (Tang et al, 2022) and Vina-GPU 2.0. AutoDock Vina considers the target protein as rigid and only the ligand is flexible. Docking of a ligand-receptor pair took on average ~3 s on 1 GPU and 1 CPU core. Since even with Vina-GPU, the docking process for a large number of molecules (around 1,350,000 in the first iteration and around 450,000 in the subsequent iterations) is still computationally expensive, this process was parallelized, and compounds were docked in batches of 1000. The outputs of these independent processes were then combined for the next phases of the pipeline. The code repository provided, however, offers a pipeline path to use AutoDock Vina instead of Vina-GPU in this step, in case GPU cores are not available.

## Downstream analysis

As the extracted number of molecules from the pipeline was still rather large (3 million), molecules were clustered, and the molecule with the best-predicted hit score of each cluster was kept. The molecules were clustered using the Taylor-Butina clustering algorithm (Butina, 1999) based on the Morgan fingerprints of size 1024 with radius 2 and Tanimoto similarity for the distance matrix, one of the most popular similarity measures for chemical substances, using Chemfp 1.x (Dalke, 2019).

For the best-predicted hit score molecule of each cluster, its conformation was acquired similarly as described in Phase 2 of the deep docking pipeline (Phase 2—Ligand preparation) and docked conventionally with Vina-GPU following a similar process to Phase 3 of the Deep Docking pipeline (Phase 3—Conventional docking) described earlier. Subsequently, the best Vina scoring 100,000 molecules in their predicted best poses were also scored with FRED (McGann, 2011). FRED is an alternative physics-based docking method with a different internal scoring function than AutoDock Vina. It takes a rigid receptor-ligand pair and provides a Chemgauss4 score for the complex. Using orthogonal docking methods, in our case AutoDock Vina and FRED, to make the selection makes the process more robust to artifacts of the individual methods (Lyu et al, 2023; Wu et al, 2024; Palacio-Rodríguez et al, 2019).

For ranking and selecting molecules based on both scores, three different methods were employed, and the overlapping best molecules formed the selected set. The first method involved taking the top 8000 molecules based on Vina and FRED scores, respectively, and retaining molecules appearing in both sets. The second method normalized Vina and FRED scores, summed them up, and then took the top-ranked molecules from the ranked list. The number of top-ranked molecules was set equal to the output of the first method, for consistency. The third method combined both methods, where we first identified top-scoring molecules both in the Vina and FRED set (thresholded on values of worst-scoring Vina and FRED molecules, respectively, from the set identified by the first method), and then we normalized scores and ranked them the same way as in the second method. Again, the number of top-ranking molecules in this method was set equal to the previous outputs for consistency.

## Final selection

As Aβ42 aggregation inhibitors should penetrate the blood-brain barrier to reach the central nervous system, we filtered the set of molecules using DeePred-BBB (Kumar et al, 2022) and MPO Guacamol (Brown et al, 2019) scores. In the final selection, molecules had to pass the DeePred-BBB filter as well as have a Guacamol MPO score >0.9 (Horne et al, 2023). From the resulting list, the final molecules were selected using visual plot-based thresholding based on more optimal Vina and FRED scores. This was done purely to reduce the number of molecules for experimental validation. From these molecules, the ones available for purchase were obtained and subsequently tested in aggregation assays.

## Experimental preparation

Recombinant Aβ42 expression. The recombinant Aβ42 peptide (MDAEFRHDSGY EVHHQKLVFF AEDVGSNKGA IIGLMVGGVV IA), here called Aβ42, was expressed in the *E. coli* BL21 Gold (DE3) strain (Stratagene, CA, USA) and purified. The purification procedure involved sonication of *E. coli* cells, dissolution of inclusion bodies in 8 M urea, and ion exchange in batch mode on diethylaminoethyl cellulose resin, followed by lyophilization. The lyophilized fractions were further purified using a Superdex 75 HR 26/60 column (GE Healthcare, Buckinghamshire, UK), and eluates were analysed using non-native SDS-PAGE for the presence of the desired peptide product. The fractions containing the recombinant peptide were combined, frozen using liquid nitrogen, and lyophilized again.

Aβ42 aggregation kinetics and fibril preparation. Solutions of monomeric Aβ42 were prepared by dissolving the lyophilized Aβ42 peptide in 6 M guanidinium hydrochloride (GuHCl) (Appendix Fig. S1) shows AKTA chromatogram confirming successful isolation of monomers). Monomeric forms were purified from potential oligomeric species and salt using a Superdex 75 10/300 GL column (GE Healthcare) at a flow rate of 0.5 mL/min, and were eluted in 20 mM sodium phosphate buffer, pH 8, supplemented with 200 μM EDTA and 0.02% $NaN_3$. The center of the peak was collected, and the peptide concentration was determined from the absorbance of the integrated peak area using $\epsilon 280 = 1490 \, \text{lmol}^{-1}\text{cm}^{-1}$. The obtained monomer was diluted with buffer to the desired concentration and supplemented with 20 μM thioflavin T (ThT) from a 2 mM stock. Each sample was then pipetted into multiple wells of a 96-well half-area, low-binding, clear-bottom, and PEG-coated plate (Corning 3881), 80 μL per well, in the absence and the presence of different molar equivalents of small molecules (1% DMSO). Assays were initiated by placing the 96-well plate at 37 ℃ under quiescent conditions in a plate reader (Fluostar Omega, Fluostar Optima or Fluostar Galaxy, BMG Labtech, Offenburg, Germany). The ThT fluorescence was measured through the bottom of the plate using a 440 nm excitation filter and a 480 nm emission filter.

Testing of compounds. All molecules were purchased from Enamine with an average purity of 97.5%. Conditions for the initial experiment were 2 μM Aβ42, 2% seed, 1% DMSO, 20 μM molecule. For the full kinetic characterization, the analysis for unseeded and 50% conditions was additionally performed and different stoichiometry explored, with 5 and 10 μM of the molecule.

Surface plasmon resonance (SPR) experiments. The Aβ42 peptide was produced as described previously (Abelein, 2020). Lyophilized Aβ42 was dissolved in 6 M GuHCl and subjected to size-exclusion chromatography on a Superdex Peptide 10/300 GL column (Cytiva). Purely monomeric Aβ42 was eluted in 20 mM sodium phosphate, 0.2 mM EDTA, pH 8.0. 20 μM monomeric Aβ42 was incubated overnight at 37 °C in quiescent conditions in a 96-well half-area non-binding plate (Corning) at 90 μL volume per well. Fibril formation was monitored using replicate wells containing 20 μM Aβ42 monomer supplemented with 20 μM ThT in a FluoSTAR Omega plate-reader (BMG Labtech).

All SPR experiments were carried out at 25 °C in a Biacore T200 instrument (GE Healthcare). The running buffer used was 20 mM sodium phosphate, 0.2 mM EDTA, pH 8.0, supplemented with 2.5% DMSO and 0.005% Tween-20. A Series S Sensor Chip CM3 (Cytiva) was primed in the running buffer and activated twice using 200 mM 1-ethyl-3-(3-dimethylaminopropyl)-carbodiimide (EDC; Invitrogen) and 50 mM N-hydroxysuccinimide (NHS; Invitrogen) at a flow rate of 10 μL/min and contact time of 420 s per round. Aβ42 fibrils were

diluted to 2 μM in 10 mM sodium acetate, pH 4.0 (Cytiva) and immobilized onto the sample flow cell at 5 μL/min in pulses of 720 s while running buffer was flowed through the reference channel. Excess reactive groups on both the channels were deactivated in two rounds using 1 M ethanolamine-HCl, pH 8.5 (Cytiva) at a flow rate of 10 μL/min and contact time of 420 s per round. The running buffer was passed over the sensor surface at 30 μL/min overnight to remove any unbound fibril. The net fibril immobilization recorded was about 1700–1800 RU (1.7–1.8 ng/mm$^2$).

A multi-cycle kinetics experiment with a contact time of 60 s (M1) or 120 s (M11) and a dissociation time of 600 s was carried out to determine the binding affinity of the drug molecules to the immobilized fibrils. Drug concentrations used were threefold dilutions of 2.16 μM (M1) or 3 μM (M11) in duplicate runs. The data recorded were normalized by double-referencing. To measure steady-state affinity, the equilibrium dissociation constant $K_D$ was determined by fitting the response recorded 4 s before the end of each injection ($R_{eq}$) against analyte concentration ($C$) to the following 1:1 binding model (Eq. 1) using GraphPad Prism

$$R_{eq} = CR_{max}/(K_D + C) + R_{max}/5 \qquad (1)$$

where $R_{max}$ is the maximum analyte response.

hiPSC culture and maintenance.   The human induced pluripotent stem cell (hiPSC) line WTSIi018-B-1 (RRID:CVCL_9S58, WT KOLF2) was used in this study. Cells were cultured on GelTrex™-coated T75/T175 flasks (#A141320, Gibco) in supplemented mTeSR Plus (#100-0276, StemCell Technologies). For passaging, hiPSC colonies were treated with ReLeSR™ (#100-0484, StemCell Technologies) for 1 min. After removing the passaging reagent, the cells were incubated at 37 °C for 2 to 3 min. The colonies were then gently washed out of the flask with mTeSR, collected in a 50-mL sterile tube, and allowed to settle for 5 min at room temperature. The supernatant was carefully aspirated, and the colonies were resuspended in mTeSR before being transferred to freshly-GelTrex-coated flasks, minimizing the pipetting steps. iPSCs were subcultured with a 1:4 to 1:6 splitting ratio.

Generation of cortical progenitors from hiPSCs.   We differentiated hiPSCs into cortical progenitor cells following a previously established protocol (Gonzalez-Diaz et al, 2024), adapted from an earlier one (Shi et al, 2012). hiPSC colonies were enzymatically dissociated to single cells using StemPro Accutase (#A1110501, Gibco) and plated at full confluence on GelTrex™-coated surfaces. Cells were cultured overnight in mTeSR supplemented with ROCK inhibitor before neural induction. For neural maintenance, we used a 1:1 mixture of N-2 and B-27 media (NMM). The N-2 medium contained DMEM/F-12 GlutaMAX, 1X N-2 (#17502048, Gibco), insulin (5 μg/mL, #I9278, Sigma), nonessential amino acids (100 μM, #11140050, Gibco), and 2-mercaptoethanol (100 μM), while B-27 medium comprised Neurobasal (#21103049, Gibco) with 1X B-27 (#17504044, Gibco). To induce neuroepithelial formation, cells were cultured for 10–12 days in NMM containing SB431542 (10 μM) and LDN-193189 (100 nM), with daily medium changes. The resulting neuroepithelium was treated with PluriSTEM™ Dispase-II Solution (#SCM133, Sigma-Aldrich), manually broken into 100–200 μm fragments, and replated on GelTrex™-coated surfaces in NMM.

Upon rosette formation (days 15–16), cultures were treated with CHIR-99021 (1 μg/mL) and FGF2 (20 ng/mL) for 4 days, followed by a daily treatment with CHIR-99021 alone until cryopreservation at days 19–22.

Differentiation of cortical NPCs to glutamatergic neurons.   Cortical NPCs were thawed on GelTrex™-coated six-well plates in NMM at 100% confluency. The medium was refreshed for 3 consecutive days. At day 4, neural progenitors were collected with the use of Accutase (#A6964, Sigma-Aldrich) and dislodged into single cells using NMM enriched with ROCK inhibitor (#SCM075, Sigma-Aldrich) and 100 U/mL of penicillin and streptomycin (#15140122, Gibco). Cell suspension was then distributed into PerkinElmer CellCarrier Ultra 96-well plates precoated with 0.002% poly-L-ornithine (#P4957, Sigma-Aldrich) and 10 μg/mL of laminin (#L2020, Sigma-Aldrich), at a density of 90–100k cells/well. The culture medium was refreshed with NMM supplemented with 10 ng/mL of rhBDNF (#248-BDB, R&D Systems), 10 ng/mL of rhGDNF (#212-GD, R&D Systems), 200 μM of L-ascorbic acid (#A4544, Sigma), 500 μM of cAMP (#D0627, Sigma), 1X of CultureOne (#A3320201, Gibco), and 100 U/mL of penicillin-streptomycin (Pen/Strep) on days 1, 4, and 7 post-seeding. Cells were also exposed to 10 μM DAPT (#2634/10, Tocris) on days 1 and 4. Starting on day 11, the medium was changed every two to three days using BrainPhys™ Neuronal Medium (#05790, StemCell Technologies) supplemented with 1X N-2, 1X SM1 (#05711, StemCell Technologies), 10 ng/mL of BDNF (#248-BDB, R&D Systems), 10 ng/mL of GDNF (#212-GD, R&D Systems), 200 μM of L-ascorbic acid (#A4544, Sigma), 500 μM dibutyryl cyclic-AMP (#D0627, Sigma-Aldrich), 1X CultureOne™ (#A3320201, Gibco), and 1X Pen/Strep.

Aβ42 treatment of glutamatergic neurons.   Recombinant Aβ42 was expressed and purified, and monomers and fibrils for cell treatment were prepared as reported previously (Díaz et al, 2024). Importantly, monomers were freshly purified before each treatment. Fibrils were stored at RT in Protein LoBind Tubes and used less than one week after their preparation. At DIV24, the full medium was removed and 100 μL of supplemented BrainPhys medium was added per well. Cells were then treated with a solution of 100 μL of Aβ42 monomer, preformed Aβ42 fibrils, and drug/DMSO. The treatment solutions were prepared as follows. Preformed Aβ42 fibrils and drug/DMSO were combined to the desired concentrations in a supplemented medium and then carefully mixed 20 times with the use of a pipette. The mixture was incubated at room temperature for 1 h. Afterwards, freshly purified Aβ42 monomers were added to the fibrils and drug mixtures to the desired concentration. The solution was then homogenized by pipetting eight times right before treatment. 20 mM NaP buffer (pH = 8, 0 mM EDTA) was used as a vehicle. Final concentrations were as follows: 500 nM Aβ42 monomers, 50 nM Aβ42 fibrils, and 2 μM of either M1 or M11 compounds. The vehicle or protein mixture comprised no more than 6% v/v of the total volume after cell treatment, while drugs/DMSO were limited to 0.1% v/v. The second treatment was prepared as explained above. After 48 h, the supernatant was collected, and the cells were fixed for immunostaining.

Immunocytochemistry.   Neurons were fixed with 4% PFA in D-PBS (+/+) for 15–20 min at RT. The fixation solution was rinsed three times with D-PBS (+/+). Neurons were then permeabilized with 0.1% Triton™ X-100 (#85111, Thermo Scientific) for 30 min at room

temperature. Cells were then incubated for at least 1 h at room temperature in immunofluorescence blocking buffer (#12411, Cell Signaling Technology). Afterwards, blocking solution was removed, and cells were incubated with a 1:500 dilution of the APP/ Aβ42 specific WO2 antibody (#MABN10, Sigma-Aldrich) for 16 h at 4 °C. Cells were then washed three times with D-PBS (+/+) followed by incubation with AlexaFluor555 anti-mouse (#A-21422, ThermoFisher) secondary antibody, kept at a 1:1000 dilution ratio in blocking solution. Cells were rinsed three times with D-PBS (+/+) and stained with Hoechst 33342 (#H3570, ThermoFisher) before imaging. Plates were imaged using the Opera Phenix High-Content Confocal microscope at a magnification of 20X. Images of 29 unique fields of view per technical replicate ($n = 3$) were obtained for quantification. Image analysis and quantification were performed with the Harmony High-Content Imaging and Analysis Software (PerkinElmer).

### Caveats

Although the method of downloading SDFs directly from ZINC20 offers significant advantages compared to traditional conformer generation methods, it comes with some caveats. The first caveat, the SDF files downloaded from ZINC20 do not necessarily match the SMILES of the database. Hence, the Morgan fingerprints are corrected to match the downloaded compounds before being used for training the model. We note, however, that there could be issues with ZINC ID queried, downloaded search-page compounds, especially for charged molecules (related to missing hydrogens). Despite the protocol working well for our use-case and the code being adjusted to work around these issues for future runs, the user can use OpenEye's Tautomers tool (around 48 seconds/200 SDF molecules with mean molecular weight 334 Da on 1 core) to correct the conformations or do open-source generation with RDKit using the same low-energy conformation generation method as this protocol, alternative fast RDKit generation, or the OpenEye's OMEGA tool (around 54 s/200 SDF molecules with mean molecular weight 334 Da on 1 core) the original protocol uses instead. OpenEye tools are not open-source, although a free license can be obtained for academic purposes. Simultaneously, ZINC20 does not provide 3D conformations for certain compounds and thus, only their 2D conformation is available, which might produce inaccurate results in the docking procedure. Therefore, in our downstream analysis, we removed the compounds associated with 2D input conformations before proceeding to rank and select the compounds. For future iterations, we adjusted the code so that in every step of the pipeline, the number of compounds for the training set augmentation is oversampled by an additional 20%, and the final set of compounds is randomly selected from the compounds that have 3D conformations available. For downstream analysis in future iterations, the compounds without 3D conformations are excluded after the ligand preparation and before the Vina docking step. As an alternative solution, again, missing 3D conformations could be generated with RDKit. However, in an experiment with a random sample of 1000 molecules with missing 3D conformations in ZINC20, RDKit managed to generate conformations only for 28% of them, suggesting that these compounds might be challenging to generate conformations for, and hence, the method of oversampling is a better option. Lastly, we used OpenBabel for ligand format conversions. An alternative to this is to replace this step with Meeko (Forli Lab, 2024), which can

be used to prepare the ligands as well as convert the Vina outputs back to SDF format, and in some cases, seems to be a more seamless and accurate option.

## Data availability

The open-source deep docking pipeline is freely available on GitHub at https://github.com/MichaelaBrezinova/open_source_deep_docking_protocol. The codebase contains also standalone scripts (for example, for filtering datasets, batch docking, etc.) that can be useful outside of the scope of this project. Standalone scripts that are not from us (for example, the Taylor-Butina clustering algorithm and RDKit confirmation generation) have their authors referenced within the scripts themselves.

The source data of this paper are collected in the following database record: biostudies:S-SCDT-10_1038-S44320-025-00159-5.

## Peer review information

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

## Acknowledgements

This work was supported by UKRI grants 10059436, 10061100, and 10138075.

## Author contributions

**Michaela Brezinova**: Conceptualization; Resources; Data curation; Software; Formal analysis; Validation; Investigation; Visualization; Methodology; Writing—original draft. **Z Faidon Brotzakis**: Conceptualization; Resources; Data curation; Software; Formal analysis; Validation; Investigation; Visualization; Methodology; Writing—review and editing. **Robert I Horne**: Formal analysis; Investigation; Methodology; Writing—review and editing. **Vaidehi Roy Chowdhury**: Resources; Data curation; Formal analysis; Investigation; Visualization; Methodology; Writing—review and editing. **Rebecca C Gregory**: Investigation; Methodology. **Yuqi Bian**: Resources; Data curation; Formal analysis; Investigation; Visualization; Methodology. **Alicia González Díaz**: Conceptualization; Resources; Data curation; Formal analysis; Supervision; Validation; Investigation; Visualization; Methodology; Writing—review and editing. **Francesco Gentile**: Conceptualization; Resources; Data curation; Software; Formal analysis; Supervision; Validation; Investigation; Methodology; Writing—review and editing. **Michele Vendruscolo**: Conceptualization; Resources; Data curation; Software; Formal analysis; Supervision; Funding acquisition; Validation; Investigation; Visualization; Methodology; Writing—original draft; Project administration.

Source data underlying figure panels in this paper may have individual authorship assigned. Where available, figure panel/source data authorship is listed in the following database record: biostudies:S-SCDT-10_1038-S44320-025-00159-5.

## Disclosure and competing interests statement

The authors declare no competing interests.

# Expanded View Figures

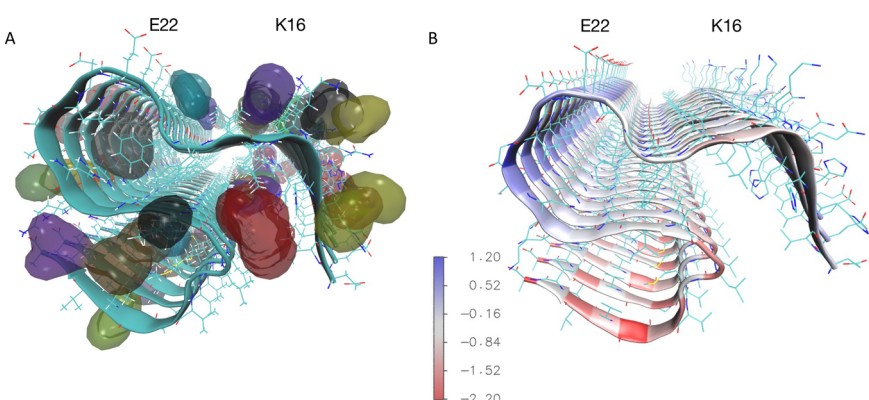

**Figure EV1.   Selection of the binding site on the amyloid fibril structure of Aβ42.**

The structure of the amyloid fibril of Aβ42 used in this work (PDB 2MXU) is depicted in a cartoon and line representation. (**A**) Prediction of binding sites using Fpocket (Le Guilloux et al, 2009), colored with different volumes. The selected binding site comprises residues [16]KVFAHLE[22]. (**B**) Solubility predictions scores per residue from CamSol (Sormanni et al, 2015). The color code ranges from red, corresponding to low solubility, to blue corresponding to high solubility. The selected binding site involves residues of low to medium solubility.

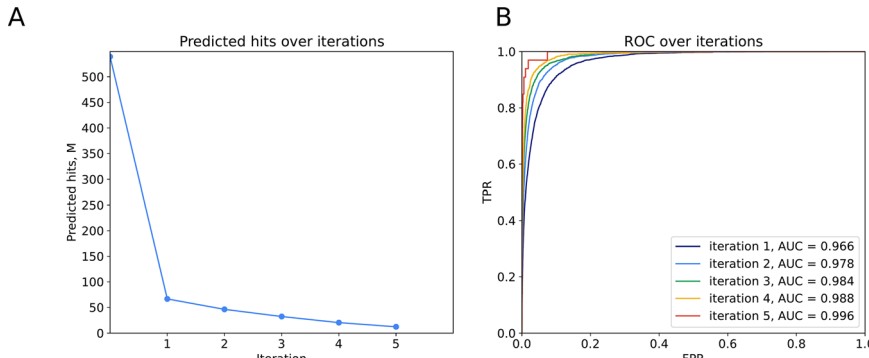

**Figure EV2.  Iterative implementation of the deep docking pipeline.**

(**A**) Number of molecules (in millions, M) predicted to be hits as a function of the iteration. In the iterative process, we reduced the initial library size of 539 million to 12 million predicted hits. (**B**) ROC curves and increasing AUC values over iterations (FPR false-positive rate, TPR true-positive rate)

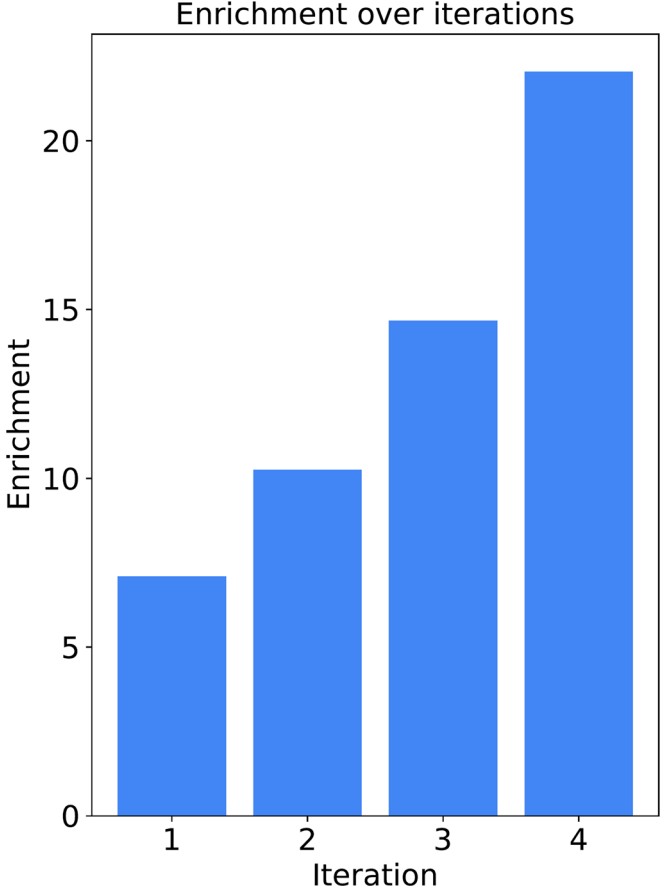

**Figure EV3. The enrichment of a random sample of model-predicted hits increases with iterations.**

The enrichment for iteration $n$ was calculated using $E_n = \frac{TP_{n,n}}{TP_{n,0}}$, where $TP_{n,n}$ are virtual true positives in a random sample of predicted hits generated by the best model of iteration $n$ (in our case, this random sample is the sample used to enrich the training set for iteration $n+1$) given model-established hit threshold of iteration $n$. $TP_{n,0}$ represent virtual true positives in the initial random sample of the original library (in our case, the training set of the first iteration), given the model-established hit threshold of iteration $n$. The virtual true positives in both cases mean molecules of which the Vina score passed the respective threshold.

A (M1)                                        B (M11)

C (Adapalene)                                 D (Thioflavin T)

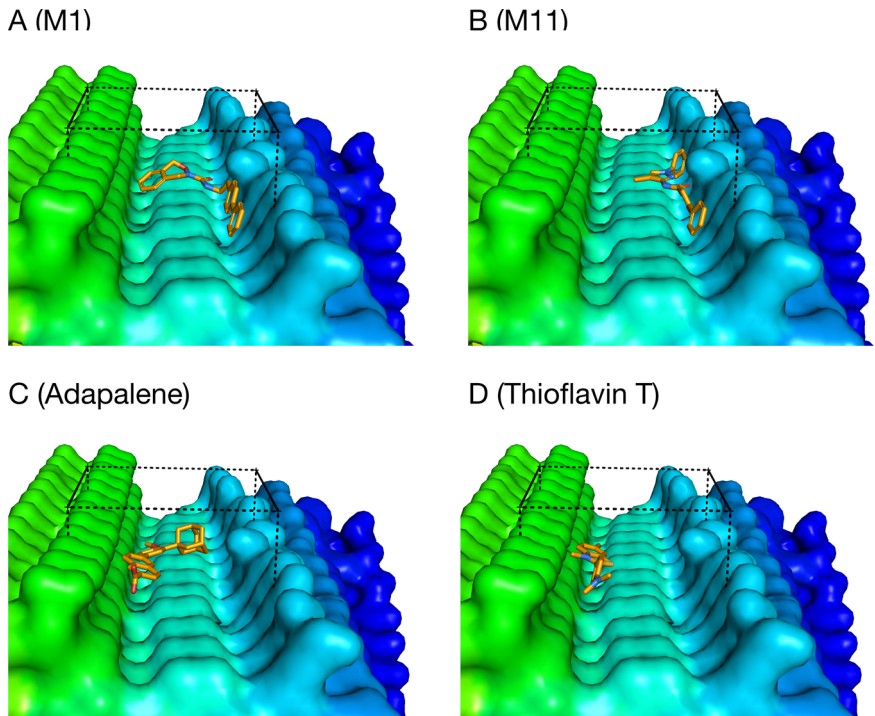

**Figure EV4.    Predicted best docking poses of the compounds discussed in this work to Aβ42 fibrils within the binding site (black dashed line).**

(**A**, **B**) M1 (**A**) and M11 (**B**) are the two most potent hits. (**C**, **D**) Predicted docking poses for adapalene (**C**) and thioflavin T (**D**) are also shown for comparison. Panels are created using PyMOL (Schrödinger LLC, 2021).

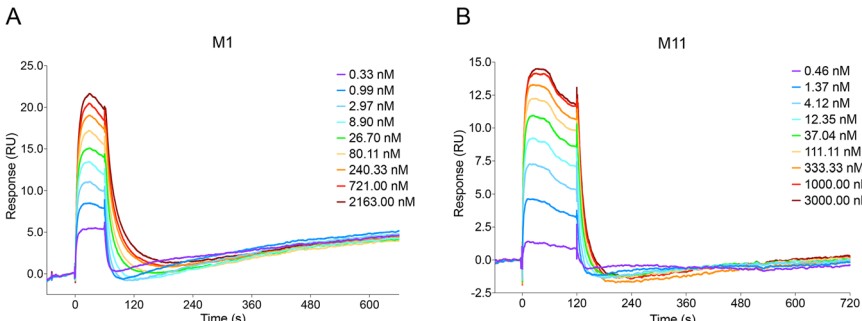

**Figure EV5.  SPR sensorgram of binding kinetics of drug molecules to Aβ42 fibrils.**

The SPR response of binding of M1 (**A**) or M11 (**B**) at different concentrations to Aβ42 fibrils immobilized at 1700–1800 RU onto CM3 sensor chip (Cytiva) are shown after removal of outlier replicates ($n = 2$).

