## [Peer Review File · Molecular Systems Biology]

Identification of high-affinity secondary nucleation inhibitors of A β aggregation using Deep Docking

Michaela Brezinova, Z. Brotzakis, Robert Horne, Vaidehi Roy Chowdhury, Rebecca Gregory, Yuqi Bian, Alicia Gonzalez-Diaz, Francesco Gentile, and Michele Vendruscolo

Corresponding author(s): Michele Vendruscolo (mv245@cam.ac.uk)

Review Timeline:

Submission Date:	25th May 25
Editorial Decision:	27th May 25
Revision Received:	9th Aug 25
Editorial Decision:	18th Aug 25
Revision Received:	1st Sep 25
Accepted:	24th Sep 25

Editor: Jingyi Hou

Transaction Report:

This manuscript was transferred to Molecular Systems Biology following peer review at another journal.

Reviewer #1:

The authors have adequately addressed all my questions. I have no further comments and recommend the paper for publication.

We thank the reviewer for the positive assessment of our work.

Reviewer #2:

Thank the authors for their efforts in addressing some of my concerns. I would appreciate it if authors could indicate what they changed in the text directly under each review comment in their rebuttal letter. This would be more efficient than requiring reviewers to locate the corresponding changes within all the highlighted text. I am concerned that I may have overlooked some of the text that have been added to address my concerns.

We apologise for the lack of clarity in our original response. We provide here further explanations, which we hope the reviewer will find helpful.

Regarding the reply, 'In the revised version of the manuscript, we have now commented on the fact that ThT binds many different forms of amyloid fibrils, with different binding modes,' I found only the following: 'This is consistent with past observations that adapalene inhibits secondary nucleation without binding to the amyloid fibrils. We also note that ThT is a potent binder of amyloid fibrils in various binding modes, but it does not act as an aggregation inhibitor.' This corresponds to the reply in the rebuttal letter. I apologize if I missed anything else that also relates to this point. First of all, there are no literature references cited in this newly added text, especially to ease my concern that the stoichiometry between inhibitors and A β 42 fibril may be 1:1. This also doesn't help justify the pose predictions of the docking results in Supplementary Fig. 4. Secondly, if the statement that adapalene inhibits secondary nucleation without binding to the amyloid fibrils is true, does that falsify the docked poses shown in Supplementary Figure 4, because the docking in this study does show potential binding of adapalene to amyloid fibrils?

We thank the reviewer for this detailed and thoughtful comment, which gives us the opportunity to clarify several important points in our study.

First, regarding the statement that ThT binds to amyloid fibrils in various modes, we have now added explicit literature references in the revised manuscript to support this claim. In particular, we cite the reviews by Biancalana and Koide (Biochim. Biophys. Acta, 2010) and by Groenning (J. Chem. Biol. 2010), which describe how ThT typically binds by being oriented along the axis of amyloid fibrils. These citations should help clarify the structural diversity of ThT binding and its lack of aggregation inhibition.

‘Adapalene is an inhibitor of A β 42 aggregation [49], however, via different mechanisms than fibril binding. M1 and M11 exhibit similar docking poses (Appendix, Supplementary Figure 4A,B), which differ from those of adapalene and ThT, as they bind a different groove on the fibril surface (Appendix, Supplementary Figure 4C,D). This is consistent with past observations that adapalene inhibits secondary nucleation through a mechanism that does not involve binding the amyloid fibrils [49]. This inhibition likely takes place through interactions with A β monomers or oligomeric intermediates, although this mechanism is typically less effective than direct inhibition at the fibril surface [37,49]. We also note that ThT is a potent binder of amyloid fibrils in various binding modes, but it does not act as an aggregation inhibitor [50,51].’

Second, we would like to address the reviewer’s concern regarding the docking poses of adapalene. While our docking results in Figure S4 suggest that adapalene can bind to amyloid fibrils, it does not bind to the specific site on the fibril surface responsible for secondary nucleation. This is a crucial distinction: binding to a fibril does not necessarily imply inhibition, since inhibition of secondary nucleation requires occupation of a specific functional site. This is the site targeted by our designed compounds M1 and M11, as shown in Figure S4. From this figure, one can see that M1 and M11 bind a different groove on the fibril structure than ThT and adapalene. We have clarified this point in the revised manuscript.

Moreover, adapalene may inhibit secondary nucleation by interacting with A β monomers or oligomeric intermediates, although this mechanism is typically less effective than direct inhibition at the fibril surface. We now state this explicitly in the manuscript (see sentence in red above) to reconcile the apparent discrepancy between docking results and experimental observations.

Lastly, regarding the cryo-EM structures (PDB IDs: 7YNM and 7YNN) that suggest a 1:1 stoichiometry of ThT binding, we note that these structures display ThT binding perpendicularly to the fibril axis—unlike the longitudinal binding mode more commonly observed and reported in the literature (see the two references above). While these newer structures offer valuable insights, they are unpublished at the time of writing. The longitudinal mode is typically associated with non-stoichiometric binding, which better reflects the scenario observed in our study.

We hope these clarifications help address the reviewer’s concerns.

In terms of the novelty part, the authors emphasize that the novelty lies in the report of the first selective and potent inhibitors for A β 42 aggregation. I am not an expert in this domain, so I will ask the other reviewers and the editor to weigh in on that part. Despite all the clarifications and changes related to the computational part in the current text, I am sticking to my original comments that the novelty of the computational part is mild. It seems more like an application story of a previously demonstrated method across various protein targets.

We appreciate the request by the reviewer of providing a more compelling explanation for the novelty of our study.

By building upon the previously published Deep Docking protocol, our study introduces significant methodological enhancements aimed at improving accessibility, efficiency, and practical application of the deep learning-based docking approach. Specifically:

Open-Source Implementation: Unlike the original method that required proprietary software, we have developed a fully open-source pipeline by integrating freely available software, such as AutoDock Vina and RDKit. This important step removes licensing barriers, significantly broadening accessibility and reproducibility for the wider scientific community.

Computational Efficiency: We have markedly reduced the computational demands associated with ligand preparation by utilizing publicly available pre-generated ligand conformations from ZINC20. This approach provides substantial computational savings compared to generating conformers de novo, greatly accelerating the pipeline.

GPU Acceleration: We integrated GPU-accelerated docking via Vina-GPU, which offers substantial time efficiency, enabling the rapid virtual screening of ultra-large chemical libraries (hundreds of millions of compounds) within a practically manageable timeframe.

Robust Experimental Validation: Unlike the original protocol, our pipeline includes extensive downstream analysis with multiple layers of compound filtering, rigorous experimental validation through kinetic assays, surface plasmon resonance (SPR) measurements, and validation in biologically relevant human neuronal models. This robust experimental framework clearly demonstrates the practical utility and biological relevance of our computational innovations.

Furthermore, our work demonstrates the impact of our application of machine learning to clinical translation. Alzheimer's disease is still largely incurable, with over 400 failed clinical trials. The only 3 disease-modifying drugs approved to date by the FDA are all antibodies targeting A β aggregates. This is remarkable, since Alzheimer's disease is a complex disorder, and dozens of other possible therapeutic targets have been intensely studied. In essence, the 3 approved antibodies provide a strong validation of the amyloid hypothesis, according to which targeting A β aggregates is the key to treating this disease.

However, these antibodies are not widely used since clinicians are wary of the potentially fatal side effects of these treatments. The primary problem is that antibodies are not part of the brain immune system, and their presence triggers a series of acute pathological reactions.

Our machine learning approach provides for the first time a route to finding a solution to this problem. We show how to replace antibodies with small molecules with the same mechanism

of action. The change of the compound class is crucial, since small molecules can enter the brain and engage their targets without causing off-target effects.

For over three decades, the search of small molecules with this mechanism of actions has defeated researchers. As we show in our work, the advent of machine learning methods has enabled the screening of chemical libraries of hundreds of millions of compounds, making it possible to identify potent therapeutic candidates.

In summary, I do not think the rebuttal letter is strong enough to justify publication in this journal.

We hope that our further explanation will be helpful to understand the impact of our work.

Reviewer #3:

The authors have significantly enhanced the manuscript by including a benchmarking section in the Introduction, where they compare their Deep Docking pipeline to traditional docking methods. This addition provides valuable context and highlights the advantages of their approach. They have clarified the criteria used for selecting the 35 compounds for experimental validation. They also expanded the experimental validation results, presenting detailed quantitative data on the potency and binding affinities of the inhibitors. In addressing potential off-target effects, the authors provided supporting data from a human neuronal model, which confirmed the selectivity of the identified inhibitors. They added the discussion about the limitations of their approach, including potential biases in the ZINC20 library and limitations of the docking algorithms. Additionally, they explored the broader applicability of their methodology to other diseases. Overall, I think their responses and revisions have significantly enhanced the quality and robustness of the study.

We are grateful to the reviewer for the acknowledging that our revisions have significantly enhanced the quality and robustness of the study.

27th May 2025

Manuscript Number: MSB-2025-13140-T

Title: Identification of high-affinity secondary nucleation inhibitors of A β aggregation using Deep Docking

Author: Michaela Brezinova

Z. Brotzakis

Robert Horne

Vaidehi Roy Chowdhury

Rebecca Gregory

Yuqi Bian

Alicia Gonzalez-Diaz

Francesco Gentile

Michele Vendruscolo

Dear Michele,

Thank you for submitting your work to Molecular Systems Biology. We have now reviewed the revised manuscript, along with your detailed point-by-point responses to the reviewers' comments from the previous journal. Overall, we believe your study makes a valuable contribution to the field, and we are pleased to inform you that it will be accepted for publication pending minor revisions, as outlined below.

Based on your responses, we are satisfied that the technical issues and related concerns have been adequately addressed. In particular, with regard to Reviewer #2's comments on the overall novelty of the study, we find that your clarification and justification in the response are clear and convincing.

At a more editorial level, please address the following issues:

- Please provide a .docx formatted (rather than PDF) version of the manuscript text (including legends for main figures, EV figures and tables).
- Please provide individual production quality figure files as .eps, .tif, .jpg (one file per figure) and remove the figures from the manuscript file.
- Please provide a .docx formatted letter INCLUDING the reviewers' reports and your detailed point-by-point responses to their comments. As part of the EMBO Press transparent editorial process, the point-by-point response is part of the Review Process File (RPF), which will be published alongside your paper.
- Please note that all corresponding authors are required to supply an ORCID ID for their name upon submission of a revised manuscript.
- We replaced Supplementary Information with Expanded View (EV) Figures and Tables that are collapsible/expandable online (see examples in <http://msb.embopress.org/content/11/6/812>). A maximum of 5 EV Figures can be typeset. EV Figures should be cited as 'Figure EV1, Figure EV2' etc... in the text and their respective legends should be included in the main text after the legends of regular figures.

Additional Tables/Datasets should be labeled and referred to as Table EV1, Dataset EV1, etc. Legends have to be provided in a separate tab in case of .xls files. Alternatively, the legend can be supplied as a separate text file (README) and zipped together with the Table/Dataset file.

For the figures and tables that you do NOT wish to display as Expanded View figures, they should be bundled together with their legends in a single PDF file called *Appendix*, which should start with a short Table of Content. Each legend should be below the corresponding Figure/Table in the Appendix. Appendix figures and tables should be referred to in the main text as: "Appendix Figure S1, Appendix Figure S2, Appendix Table S1" etc. Please submit the Appendix as a separate PDF file.

See detailed instructions regarding expanded view here:

<https://www.embopress.org/page/journal/17444292/authorguide#expandedview>.

- Before submitting your revision, primary datasets (and computer code, where appropriate) produced in this study need to be deposited in an appropriate public database (see <http://msb.embopress.org/authorguide-dataavailability>

<https://www.embopress.org/page/journal/17444292/authorguide#dataavailability>).

The accession numbers and database should be listed in a formal "Data Availability" section (placed after Materials & Method)

that follows the model below (see also <https://www.embopress.org/page/journal/17444292/authorguide#dataavailability>). Please note that the Data Availability Section is restricted to new primary data that are part of this study.

Data availability

- RNA-Seq data: Gene Expression Omnibus GSE46843 (<https://www.ncbi.nlm.nih.gov/geo/query/acc.cgi?acc=GSE46843>)

- [data type]: [name of the resource] [accession number/identifier/doi] ([URL or identifiers.org/DATABASE:ACCESSION])

-At EMBO Press we ask authors to provide source data for the main figures. Our source data coordinator will contact you to discuss which figure panels we would need source data for and will also provide you with helpful tips on how to upload and organize the files.

- Our journal encourages inclusion of *data citations in the reference list* to directly cite datasets that were re-used and obtained from public databases. Data citations in the article text are distinct from normal bibliographical citations and should directly link to the database records from which the data can be accessed. In the main text, data citations are formatted as follows: "Data ref: Smith et al, 2001". In the Reference list, data citations must be labeled with "[DATASET]". A data reference must provide the database name, accession number/identifiers and a resolvable link to the landing page from which the data can be accessed at the end of the reference. Further instructions are available at .

- We updated our journal's competing interests policy in January 2022 and request authors to consider both actual and perceived competing interests. Please review the policy <https://www.embopress.org/competing-interests> and update your competing interests if necessary.

Please use the heading "Disclosure statement and competing interests".

- All Materials and Methods need to be described in the main text using our 'Structured Methods' format. According to this format, the Methods section includes a Reagents and Tools Table (listing key reagents, experimental models, software and relevant equipment and including their sources and relevant identifiers) followed by a Methods and Protocols section describing the methods, ideally using a step-by-step protocol format. The aim is to facilitate adoption of the methodologies across labs. Please download and fill our Reagents and Tools Table template (.docx), which you can find in our author guidelines: <https://www.embopress.org/page/journal/17444292/authorguide#structuredmethods>.

-Regarding data quantification:

Please ensure to specify the name of the statistical test used to generate error bars and P values, the number (n) of independent experiments (please specify technical or biological replicates) underlying each data point and the test used to calculate p-values in each figure legend. Discussion of statistical methodology can be reported in the materials and methods section, but figure legends should contain a basic description of n, P and the test applied.

Graphs must include a description of the bars and the error bars (s.d., s.e.m.).

-The references need to be formatted according to the Molecular Systems Biology reference style. Please list up to 10 co-authors of a paper before adding et al. in the reference list. Citations should be listed in alphabetical order.

- Please provide a "standfirst text" summarizing the study in one or two sentences (approximately 250 characters, including space), three to four "bullet points" highlighting the main findings and a "synopsis image" (550px width and 400-600 px height, PNG format) to highlight the paper on our homepage.

Here are a couple of examples:

<https://www.embopress.org/doi/10.15252/msb.20199356>

<https://www.embopress.org/doi/10.15252/msb.20209475>

<https://www.embopress.org/doi/10.15252/msb.209495>

When you resubmit your manuscript, please download our CHECKLIST (<https://www.embopress.org/pb-assets/embo-site/EMBO%20Press%20Author%20Checklist-1642513524327.xlsx>) and include the completed form in your submission.

Please note that the Author Checklist will be published alongside the paper as part of the transparent process (<https://www.embopress.org/page/journal/17444292/authorguide#transparentprocess>).

Sincerely,

Jingyi

Jingyi Hou, PhD
Senior Editor
Molecular Systems Biology

If you do choose to resubmit, please click on the link below to submit the revision online before 26 June 2025.

*** PLEASE NOTE *** As part of the EMBO Press transparent editorial process initiative (see our Editorial at <https://dx.doi.org/10.1038/msb.2010.72> , Molecular Systems Biology will publish online a Review Process File to accompany accepted manuscripts. When preparing your letter of response, please be aware that in the event of acceptance, your cover letter/point-by-point document will be included as part of this File, which will be available to the scientific community. More information about this initiative is available in our Instructions to Authors. If you have any questions about this initiative, please contact the editorial office (msb@embo.org).

The authors addressed the editorial issues.

18th Aug 2025

Manuscript Number: MSB-2025-13140R

Title: Identification of high-affinity secondary nucleation inhibitors of A β aggregation using Deep Docking

Author: Michaela Brezinova

Z. Brotzakis

Robert Horne

Vaidehi Roy Chowdhury

Rebecca Gregory

Yuqi Bian

Alicia Gonzalez-Diaz

Francesco Gentile

Michele Vendruscolo

Dear Dr. Vendruscolo,

Thank you for sending us your revised manuscript. Before we can formally accept it for publication, please address the following remaining editorial level issues:

1. Please include up to five keywords in the manuscript.
2. Add the corresponding author's email address on the title page.
3. Rename the section titled "Disclosure statement and competing interests" to "DISCLOSURE AND COMPETING INTERESTS STATEMENT."
4. Remove the "Authors Contribution" section from the manuscript.
5. Add the missing figure callouts for Figures 3A-C, 3F, and 4A-B in the text.
6. Please provide a "standfirst text" summarizing the study in one or two sentences (approximately 250 characters, including space), three to four "bullet points" highlighting the main findings to highlight the paper on our homepage.

Here are a couple of examples:

<https://www.embopress.org/doi/10.15252/msb.20199356>

<https://www.embopress.org/doi/10.15252/msb.20209475>

<https://www.embopress.org/doi/10.15252/msb.209495>

7. Reagent and Tool table should be uploaded as a separate file using the provided template :
https://www.embopress.org/pb%2Dassets/embo-site/Reagents_Tools_Table_TEMPLATE.docx

8. Address the following issues in figure legends:

- Please note that information related to n is missing in the legends of figures 5A, B
- Please note that the error bars are not defined in the legends of figures 5A, B.

9. Appendix

- Figures and tables should be listed in the Table of Contents using the format "Appendix Figure/Table Sx", and the same labeling should be consistently applied throughout the Appendix PDF.
- Appendix Table S2 should be reoriented.

10. " Methods and Materials" should be renamed to "Methods".

11. Source data:

- Source data files should be organized using a one figure per folder structure. For example, all source data files for Figure 1 should be saved in a single folder, then compressed into a .zip file named "SD Figure 1.zip" before uploading.

- We also request a completed Source Data Checklist when submitting your revised research article.

https://www.embopress.org/pb-assets/embo-site/EMBOpress_Source_Data_Checklist-1743685524943.xlsx

12. The following email addresses appear to be invalid and are bouncing: Z. Faidon Brotzakis - fb516@cam.ac.uk; Yuqi Bian -

yb279@cam.ac.auk. Please verify and provide valid email addresses.

13. In the Author Checklist, please complete the "In which section is the information available" column for all items marked "Yes."

14. Please rename and reorder manuscript sections as follows: Title page - Abstract - Keywords - Introduction - Results - Discussion - Methods - Data Availability - Acknowledgements - Disclosure and Competing Interests Statement - References - Figure Legends - Table(s) - Expanded View Figure Legends.

Click on the link below to submit your revised paper.

Yours sincerely,
Jingyi

Jingyi Hou, PhD
Senior Editor
Molecular Systems Biology

*** PLEASE NOTE *** As part of the EMBO Press transparent editorial process initiative (see our Editorial at <https://dx.doi.org/10.1038/msb.2010.72> , Molecular Systems Biology will publish online a Review Process File to accompany accepted manuscripts. When preparing your letter of response, please be aware that in the event of acceptance, your cover letter/point-by-point document will be included as part of this File, which will be available to the scientific community. More information about this initiative is available in our Instructions to Authors. If you have any questions about this initiative, please contact the editorial office (msb@embo.org).

All editorial and formatting issues were resolved by the authors.

24th Sep 2025

Manuscript number: MSB-2025-13140RR

Title: Identification of high-affinity secondary nucleation inhibitors of A β aggregation using Deep Docking

Dear Dr. Vendruscolo,

Thank you again for sending us your revised manuscript. We are now satisfied with the modifications made and I am pleased to inform you that your paper has been accepted for publication.

Sincerely,
Jingyi

Jingyi Hou, PhD
Senior Editor
Molecular Systems Biology
